# Nutrition-Related Information on Instagram: A Content Analysis of Posts by Popular Australian Accounts

**DOI:** 10.3390/nu15102332

**Published:** 2023-05-16

**Authors:** Emily Denniss, Rebecca Lindberg, Sarah A. McNaughton

**Affiliations:** Institute for Physical Activity and Nutrition, School of Exercise and Nutrition Sciences, Deakin University, Burwood, VIC 3125, Australia; r.lindberg@deakin.edu.au (R.L.); sarah.mcnaughton@deakin.edu.au (S.A.M.)

**Keywords:** nutrition communication, social media, content analysis, nutrition information, Leximancer

## Abstract

Social media is a popular source of nutrition information and can influence food choice. Instagram is widely used in Australia, and nutrition is frequently discussed on Instagram. However, little is known about the content of nutrition information published on Instagram. The aim of this study was to examine the content of nutrition-related posts from popular Australian Instagram accounts. Australian Instagram accounts with ≥100,000 followers, that primarily posted about nutrition, were identified. All posts from included accounts, from September 2020 to September 2021, were extracted and posts about nutrition were included. Post captions were analysed using Leximancer, a content analysis software, to identify concepts and themes. Text from each theme was read to develop a description and select illustrative quotes. The final sample included 10,964 posts from 61 accounts. Five themes were identified: (1) recipes; (2) food and nutrition practices; (3) body goals; (4) food literacy and (5) cooking at home. Recipes and practical information about nutrition and food preparation are popular on Instagram. Content about weight loss and physique-related goals is also popular and nutrition-related Instagram posts frequently include marketing of supplements, food and online programs. The popularity of nutrition-related content indicates that Instagram may be a useful health-promotion setting.

## 1. Introduction

Worldwide, social media has become an increasingly popular communication medium. Fifty-nine percent of the global population are active social media users [1]. Higher rates of social media use are observed in Australia, where 81% of the population actively use social media, spending an average of almost two hours on social media per day [2]. The rise of social media has been accompanied by the rise of the “influencer”. Influencers are individuals who have amassed large numbers of followers on social media, and have the ability to influence the opinions, attitudes and behaviours of their followers [3]. The public often have high levels of trust in the influencers they follow, and advertisers use this to their advantage by paying influencers to endorse and promote products [4,5]. Brands and companies also have a strong social media presence, and their accounts often have large followings [6]. Social media influencers and brand accounts are particularly prominent on Instagram, which is one of the most widely used social media platforms [6,7]. In 2021, Instagram was the fifth most visited website in the world and the third most popular platform in Australia [1,2].

Dietary intake is a major determinant of human health and social media has the potential to influence individuals’ dietary behaviours [8]. Social media has become a popular source of information about food and nutrition [9,10]. Consumers actively seek food and nutrition information on social media and are also passively exposed to it in their social media feeds [11]. Social media’s impact on food choice is not fully understood; however, there is evidence that exposure to food- and nutrition-related social media content may influence food choice in ways that promote or threaten good health. Exploratory research has revealed that using social media for recipes and healthy eating information may help consumers select and prepare healthy meals [9,12]. Conversely, research has shown that children who are exposed to marketing of discretionary foods on social media are more likely to select the advertised products [13]. A systematic review that investigated social media use and eating behaviours found that general social media engagement was associated with body dissatisfaction, restrained eating, and overeating [14]. Interestingly, the review also found that social media use was associated with healthy food selection, further evidencing the notion that social media has the power to influence healthy and unhealthy eating behaviours [14].

Analysis of social media content can provide important insights into a range of topics relevant to public health practice such as trends in health behaviours and how particular health topics are framed [15]. Due to the significant impact eating behaviours have on human health and the influence that social media can have on eating behaviours, it is important to examine how food and nutrition is discussed on popular social media platforms. Instagram is one of the most popular social media platforms globally [1], and nutrition is one of the most commonly discussed health topics on the platform [15]. However, there is a scarcity of research about the nutrition-related information published by influential Instagram accounts. Existing research in this area has focused on selected nutrition-related topics and/or individual hashtags on Instagram, such as #cleaneating or #healthyeating [16,17]. These studies have focused on all posts or the “top posts” for a certain hashtag, rather than examining content posted by wide-reaching accounts with large numbers of followers [16,17]. One study has content-analysed nutrition-related Instagram posts by influencer accounts; however, only three accounts were included in the sample, which may limit the generalisability of the findings [18]. 

Given the potential role of social media in the nutrition information environment, there is a need to investigate and characterise the nutrition-related content posted by popular and influential Instagram accounts. Therefore, the aim of this exploratory study was to examine the content of nutrition-related posts by popular Instagram accounts, using a sample of popular Australian accounts operated by influencers and companies.

## 2. Materials and Methods

This study was a qualitative content analysis of textual data from Australian Instagram posts and followed guidelines set out by Elo et al. for improving the trustworthiness of qualitative content analysis studies [19]. Standards for Reporting Qualitative Research (SRQR) were also followed and the SRQR checklist is summarised in Appendix A [20]. The first author (ED) was involved in all components of the study, from sample selection to data analysis. ED is a social media user with a Bachelor of Food and Nutrition Science (Honours), who often engages with Instagram content about nutrition and is thus an “insider” who is familiar with the topic of investigation [21]. At the time of data collection, ED did not follow any of the included accounts, although she was familiar with a small number of them. An understanding of the language and terminology used in nutrition-related Instagram content assisted ED in her interpretation of the data. 

### 2.1. Ethics

The dataset for this study was sourced from publicly available Instagram accounts. As the data used were publicly available, approval from an ethics committee was not required. Regardless, to ensure that the research was conducted in an ethical way, all data have been de-identified to protect the identities of the individuals and brands included. The illustrative quotes used were searched in Google, Bing and Yahoo search engines to ensure that they could not be linked back to the account holders.

### 2.2. Instagram Account Sample Selection

At the time this study was conducted, it was not possible to systematically search Instagram for accounts based on location, number of followers or content topics, due to the influence of Instagram’s algorithm and restricted access to the Application Programming Interface (API). In order to identify an appropriate sample for analysis, a two-stage process was undertaken. Firstly, a list of the top 1000 Australian health and fitness Instagram influencers, published by social media marketing company StarNgage, was used and listed accounts were screened for relevance [22]. Secondly, snowballing was used to capture other relevant accounts that had not already been selected. Using the accounts selected from the initial StarNgage list, accounts mentioned in Instagram’s “suggested” tab were also screened for eligibility. 

Inclusion criteria for selection of Instagram accounts were accounts with ≥100,000 followers, the brand or account holder was based in Australia, a minimum of 100 grid posts and most recent grid post published within two weeks of the screening date and at least 25% of their posts in the last month contained nutrition-related information. For the purposes of this study, nutrition-related information was defined as: information regarding heathy eating, dietary patterns, nutrients, nutritional requirements, nutritional composition of foods, nutritional supplements, health outcomes associated with foods and dietary patterns, food safety, food ethics, cooking and recipes intended for the general public. This definition was developed based on the concepts included in Vidgen and Gallegos’ definition of food literacy [23].

Account screening was conducted independently by the first author (ED) and a research assistant (JV, named in acknowledgments), and disagreements were discussed until consensus was reached. Common reasons for disagreements included miscounting the number of posts containing nutrition-related information and disagreements about whether a post contained nutrition-related information. All screening was completed using a Google Chrome browser in incognito mode and an Instagram account created for this study to reduce any potential impact from algorithms on the accounts that were suggested. 

### 2.3. Instagram Post Sample Selection

All Instagram posts from a 12-month period, excluding story posts, published by the selected accounts were downloaded. A membership with a social media analytics company, Keyhole, was used to facilitate the download of data in a manner that complied with Instagram’s terms of service. Posts from the identified accounts were downloaded through Keyhole’s platform. All posts were screened for relevance and were included if they contained a caption or image-based text that related to one or more aspect of nutrition-related information. Posts containing videos or no relevant information were excluded. Posts were screened by the lead author and a research assistant. The first 10% of posts for each included account were independently screened by both researchers for reliability, achieving 94% agreement. Disagreements were discussed until consensus was reached. Common reasons for disagreements were discussed to improve reliability before the remaining screening was conducted. Included posts that contained images of text were flagged during the screening process and the text from images was transcribed. Each post’s engagement data (total number of likes and comments) were also downloaded via Keyhole. The median number of likes for an Instagram post is 200 and the average number of comments is 24.5; posts with engagement that exceeds these numbers are considered to have good engagement [24].

### 2.4. Analysis

Instagram accounts were coded into types of account in order to describe the sample. The researchers who conducted the screening and were immersed with the data developed categories for the included Instagram accounts. The two researchers independently categorised all accounts based on the type of content the account most regularly published. Disagreements on the account’s category were discussed until a consensus decision was reached. The categories were: lifestyle influencer, fitness influencer, brand, recipes or meal ideas, nutritionist or dietitian influencer, or media entity. Instagram account categories are described in Appendix A. 

This study used a hybrid content analysis approach. In hybrid content analyses, computer-aided content analysis software and manual, investigator-led interpretations are used sequentially. Hybrid approaches are recommended in the literature for the analysis of large datasets, to overcome the shortcomings of each approach [25,26,27]. First, the computer-aided content analysis software, Leximancer, was used to content analyse the text from included Instagram posts. The software automates the content analysis of textual data through the use of machine learning. Leximancer has demonstrated to be a valid and reliable method of conducting content analysis [28,29]. The automated processing allows for the efficient analysis of large amounts of textual data, that would otherwise be highly time and resource intensive [28,29]. Leximancer may also minimise researcher bias as it determines concepts and themes using objective approaches and reduces the potential impact of subjectivity if the analysis was conducted manually [28,29]. 

Leximancer identifies “concepts”, which are groups of words that tend to travel together in text, and a thesaurus of terms is generated for each concept [30]. The co-occurrence of concepts within the text is also measured to determine the connectivity between identified concepts. The relationship between concepts is illustrated on the concept map, where concepts that are often mentioned together in the text (i.e., concepts with high connectivity) are situated close together on the map, and concepts that are rarely or not at all mentioned together are situated far apart on the concept map. Highly connected concepts are grouped into “themes”, illustrated by coloured circles on the concept map. Themes are heat mapped, whereby more important themes (i.e., themes with concepts that are more frequent in the text) are shown in warmer colours, and less important themes are shown in cooler colours. The concepts, themes and concept map are generated with Leximancer’s machine learning algorithm and the researchers are able to alter Leximancer’s text-processing settings based on the research questions, to modulate the output [30]. 

Captions and transcribed text from included posts were imported into Leximancer. An initial exploratory concept map was generated and used to inform the final text-processing settings, as outlined in Leximancer’s user guide [30]. The number of sentences analysed per block was set at four, instead of the default of two. This approach has been taken previously in social media research that has used Leximancer, because social media content is often fragmented and uses short sentences [31]. The break-at-paragraph feature was turned off because it was common for posts to separate single sentences with a paragraph or list items as dot points. The stop word “eat” was switched off to ensure it was captured in the analysis. Leximancer identified concepts that were not related to the aim of this study were removed (e.g., “link”, “bio”, “click”) to prevent the concept map being distorted toward irrelevant concepts. Similar concepts that sat close together on the exploratory concept map were combined, for example, “vitamin” and “vitamins”. Many concepts appeared very close together on the concept map and were thus illegible. The map was therefore manually edited to depict the name and position of all concepts, using an interactive feature in Leximancer that shows the positioning of each concept. Manual edits to enhance the readability of the concept map were purely visual in nature and did not impact the underlying results.

Leximancer includes an option to alter the size of the themes that appear on the concept map, where a smaller theme size includes a narrower group of concepts, and a larger theme size includes a broader group of concepts. To select a theme size, the lead author explored the concept map, auto-generated themes and their associated text segments at various theme sizes and developed a coding framework for each theme size. The coding framework was circulated to all authors who met to discuss and form a consensus decision about which theme size best represented the data. After a theme size was agreed upon, the lead author read text segments for each theme to develop descriptions for the themes, rename themes and select illustrative quotes. Due to the large volume of data, it was not feasible to read all text segments for each theme. Text segments tagged with a greater number of the theme’s concepts are reported earlier in the list, and therefore, the first 10% of text segments for each theme were read and interpreted due to their greater relevance. A random 5% of text segments for each theme were also read by the lead author to ensure that the interpretation of data was comprehensive. Text segments, theme descriptions and illustrative quotes were read by all authors to establish if there was consistent interpretation of the data between the research team. Disagreements and inconsistencies were discussed, and changes were made as necessary.

## 3. Results

From 1786 Instagram accounts that were screened, 61 accounts were identified as eligible for inclusion. A total of 22,438 posts were downloaded for the period of 22 September 2020 to 22 September 2021 and 10,964 posts were included in the final sample (Figure 1). Account categories, follower count, engagement, and number of posts for included accounts are presented in Table 1. At a theme size of 35%, 5 themes and 97 concepts were identified by Leximancer (Table 2). In order of decreasing importance, the five themes that were automatically generated by Leximancer were: “ingredients”, “healthy”, “eat”, “recipe” and “home” (Figure 2). 

Hits are the number of times a segment of text has been linked to a theme. Engagement was calculated based on engagement for posts tagged with the most frequent concept for each theme. Count is the number of times a concept appeared in the sample. Relevance is the frequency of text segments that contain the concept, proportionate to the frequency of the most dominant concept.

### 3.1. Recipes

The theme with the highest ranked importance was “ingredients”, with a total of 14,289 linked text segments and an average post engagement of 1382. There was a high degree of connectivity between Leximancer identified concepts related to ingredients (for example, “ingredients”, “coconut”, “milk”, “almond” and “water”); measurements (for example, “cup” and “tsp”); and cooking methods (for example, “method”, “blend”, “mix” and “cook”). Linked text segments for this theme primarily consisted of recipes; as such, the theme name was labelled by authors as “recipes”. The analysis of the posts included in this theme found several common characteristics. Typically, the recipes are promoted as healthy, which is achieved through ingredient substitution and through the inclusion of plant-based products. Product placement was frequent and although less ubiquitous, some posts were dedicated to particular cooking or recipe topics. 

Popular recipe content was often described as “healthy”, “guilt-free” or a “healthier version” of a food. “Healthier” versions of desserts or sweets were particularly frequent. Many recipes described the need for alternatives to refined sugar, such as honey, coconut sugar, agave or maple syrup, and used coconut oil as the source of fat, for example:


*“…V GF DF RSF [vegan, gluten free, dairy free, refined sugar free] Ingredients: … 1 1/2 tbsp coconut oil 1 tbsp maple syrup 1 tsp vanilla extract Pinch of greens powder…”*
—Recipes/meal ideas, followers: 154,318

Recipes in Instagram posts commonly contained predominantly plant-based ingredients such as grains, vegetables, fruits, nuts and lentils and were regularly vegan, or provided suggestions for how to make the recipe vegan, for example:


*“Learn how to make a creamy vegan dip using cashews and veggies that everyone can enjoy! Find our Instagram exclusive recipe below…”*
—Brand, followers: 202,815

It was also common for posts to market products by including a specific product as an ingredient in a recipe. Products marketed as ingredients included items such as fruits, vegetables, pasta, and tuna, but were most frequently dietary supplements such as protein powders:


*“…TIRAMISU BREAKFAST JARS • INGREDIENTS: « 4 tbsp chia seeds « 1 cup almond milk « 2 heaped tbsp coconut yoghurt Choc Layer: « 1 serve [brand] Vegan Triple Choc Fudge Protein Powder…”*
—Brand, followers: 194,224

A smaller portion of text segments contained a description of a food and a link to the full recipe, or listed and discussed the ingredients in packaged foods.

### 3.2. Food and Nutrition Practices

The Leximancer-identified theme, “healthy”, had a total of 12,051 hits and was identified as the second most dominant theme. The average engagement for posts was 1053. There was a high degree of connectivity between concepts relating to nutrients or food (for example, “vitamin”, “collagen”, “food”, “contains” and “rich”), and concepts relating to health or the body (for example, “healthy”, “energy”, “skin”, “muscle”, “gut” and “immune”). Text segments linked to this theme discussed various food and nutrition practices and the theme name was changed to “food and nutrition practices” to reflect this. Analysis of text segments included under this theme found that, typically, posts discussed a food and nutrition practice, such as the consumption of a food, specific nutrient, dietary pattern or supplement and an associated health outcome. 

The marketing of supplements, including information about nutrients they contained and their associated health benefits, was particularly frequent. Protein powders and pre-workout products to improve muscle mass and exercise performance, and prebiotics, probiotics, greens-, reds- and collagen-powders to improve immune function, gut health and/or promote “beauty through wellness” by improving the appearance of skin, hair and nails were frequently promoted:


*“With clinically proven collagen peptides plus bioavailable Vitamin C and Zinc, [supplement name] is the most targeted formulation for hair growth, strength and overall quality—along with improving skin, nails and gut health. Shop via link in bio [brand] #collagenbeauty”*
—Brand, followers: 167,440

The healthfulness of specific foods and dietary patterns were also frequently discussed and often included information about the nutrients contained in foods and health benefits of consuming particular foods and/or dietary patterns:


*“Pomegranate has anti-oxidant, anti-viral and anti-tumour properties and is said to be a good source of vitamins, especially vitamin A, vitamin C, and vitamin E, as well as folic acid.”*
—Lifestyle influencer, followers: 123,642

More general discussions of healthy eating practices, advice about healthy eating and supplement use, and healthy recipe provision were also included in text segments, although were less frequent than information about direct links to specific health outcomes. The majority of text segments that discussed health outcomes mentioned health benefits, rather than potential negative health outcomes. There was a large amount of marketing of supplements and food products, which included links to online stores, with the concept “shop” reaching 2449 hits.

### 3.3. Body Goals

The Leximancer-identified theme, “eat”, had a total of 11,401 hits. Posts under this theme had the highest average engagement at 2276. There was a high degree of connectivity between the concepts “results”, “diet”, “weight”, “calories”, “loss”, “goals” and “challenge”; these concepts are situated in a cluster in the concept map (Figure 2). Text segments in this theme predominantly discussed nutrition’s role in achieving body- or physique-related goals such as weight loss, fat loss or the accumulation of muscle mass, and therefore, the theme was renamed as “body goals”. Analysis of this theme revealed that posts frequently discussed how to eat to achieve specific body goals or “body transformations”. Text segments included under the “body goals” theme were similar to those captured under the “food and nutrition practices” theme, in that they discussed the relationship between a nutrition behaviour and an outcome. However, the outcomes discussed under the “body goals” theme were more related to image and changes to physique, rather than health. 

Posts frequently provided information about energy and macronutrient requirements to achieve a specific result, such as eating in calorie surplus to gain muscle or calorie deficit to lose fat. Information about intake requirements was often accompanied by the promotion of an online subscription-based program for meal plans, challenges, recipes, education, coaching and/or exercise: 


*“Amazing transformation from my client [name] from Sweden. One huge mistake a lot of women tend to make is spending too much time in a deficit. A deficit is essential for a fat loss phase but if you want to build muscle and change the shape of your body, you must spend time building your physique and fueling it with maintenance/surplus calories.”*
—Fitness/coaching influencer, followers: 246,547

Discussion of one’s relationship with food or body image was often included in text segments. Some information regarding body image and disordered or emotional eating was general and referred individuals to additional resources; however, most discussions appeared alongside the promotion of weight loss or “body transformation” subscription services:


*“…consider my premium [weight loss program]… Personalised plans to help you conquer emotional eating!!!”*
—Nutritionist/dietitian, followers: 302,464

Free example recipes from meal plan subscriptions, the discussion of nutrition science research, information about infant feeding, the marketing of supplements and results from supplement use were present but less frequently mentioned throughout text segments for the “body goals” theme.

### 3.4. Food Literacy

The theme “recipe” was identified by Leximancer and accumulated 7859 hits; posts under this theme had an average engagement of 1542. Text segments under this theme mainly discussed elements of food literacy; therefore, the theme was renamed as “food literacy”. Examination of text segments under this theme revealed that posts typically discussed the practical components of preparing food, such as cooking tips and techniques, as well as describing the sensory experience of preparing and consuming different dishes. 

There were frequent discussions on cooking and the promotion of recipes that were published elsewhere, for example, websites, blogs or books. It was common for text segments to describe a recipe’s flavours and textures, its origins or cultural history and/or emphasise the pleasure of cooking and eating:


*“Depending on where you are in the world the word biscuit means different things. In America, the biscuit is a thick, savoury square similar to Australian scones… [This] recipe dots cheese and slices of soppressa throughout for a flaky biscuit that’ll have you forgetting fried chicken.”*
—TV show/magazine, followers: 253,792

Practical information about meal preparation and infant feeding was also frequent, for example, information about safely storing, freezing, or defrosting meals, and appropriate foods to introduce to infants. Practical suggestions of meals that are “family friendly”, simple and quick to prepare, and involve minimal or affordable ingredients were also very common: 


*“If you love a one-pan wonder as much as we do, you have to try our recipe for a classic chicken cacciatore. It’s the perfect weekday meal for all the family.”*
—TV show/magazine, followers: 212,301

Text segments also included advice and information about using specific cooking methods or ingredients:


*“It features specific buckwheat noodles made using buckwheat and a combination of starches giving them a delightful springy texture…”*
—TV show/magazine, followers: 253,792

### 3.5. Home

The least dominant theme identified by Leximancer was “home”, which accumulated 1047 text segment hits. On average, post engagement was 1969. This theme contained two concepts, “home” and “year”. Text segments for this theme predominantly referred to performing tasks at home, such as home-based exercise or cooking and therefore the theme name was not changed. There was often discussion of eating, cooking or exercising at home due to COVID-19 restrictions and reference to the challenging and abnormal year of 2020. Additionally, the seasonality of foods, environmentally sustainable food practices, particular times of year, such as Christmas and the festivities involving food, and the time involved in seeing results from exercise and dieting were also discussed.

## 4. Discussion

The present study used a hybrid qualitative content analysis approach to examine the content of nutrition-related posts by popular Australian brand and influencer Instagram accounts. Five major themes were identified and the analyses of text segments from each theme are summarised into three major findings. Firstly, popular Instagram accounts posted recipes and a large amount of information about nutrition and health, cooking and food literacy. Secondly, the marketing of products, including supplements, foods, meal plans and apps was prevalent throughout Instagram posts. Finally, Instagram posts included frequent discussions on weight loss and dieting to achieve image-based goals. All accounts included in the sample had a substantial following, and engagement on the included posts was also high, indicating the reach and popularity of the nutrition-related information published on Instagram. 

Instagram posts included in the sample frequently contained recipes. Research using Instagram data has found that hashtags related to recipes and cooking at home were frequently used alongside #healthyfood [17], and influencers frequently posted recipes [18]. The prominence and popularity of recipe content observed in the present study are supported by these studies. Home cooking and food preparation is associated with diet quality [32], and there is emerging evidence that consumers use social media to source recipes [12,33,34]. Recipes in the sample frequently contained fruits, vegetables, lentils and wholegrains, and were often plant-based, which suggests that consuming recipes from Instagram may facilitate healthy eating. However, many recipes purported to be healthy or healthier than a traditional alternative and replaced typical sources of fat and sugar, such as butter and white sugar, with sources that may be perceived as healthier, such as coconut oil or honey. Dickinson et al. compared clean-eating versions of well-known recipes from blogs to their traditional counterparts and found that the so-called healthier versions contained the same amount of sugar, higher amounts of fat and similar amounts of energy [35]. While Instagram may be a source of healthy recipes, consumers may also be at risk of being misled when trying to select healthy recipes from the platform. More research is needed to determine the healthfulness of recipes published on Instagram and how the public are incorporating these recipes into their diets. 

In addition to recipes, posts also contained practical information about nutrition, procuring and preparing food, and eating for health. Similarly, qualitative studies have found that practical information about cooking—including tips on batch cooking, one-pot meals, and meal planning; and information about healthy eating, including the promotion of fruit, vegetables, and wholegrains—are prevalent on nutrition blogs [36,37]. Greater nutrition knowledge is associated with diet quality [38], and it has been posited that food literacy, which encompasses procedural knowledge of planning, selection, preparation, and consumption of food, is important for supporting healthy eating [23]. Social media has been identified as a useful setting for health and nutrition promotion due to the popularity of food-related content [39,40]. There is a growing body of literature that aims to translate influencer communication techniques into strategies for social-media-based nutrition promotion [39]. The popularity of recipe, cooking and healthy eating content observed in this study supports the use of influencer techniques or government collaboration with influencers to promote nutrition on social media, for example, informational campaigns to improve nutrition knowledge and food literacy and the promotion of healthy eating through the provision of healthy recipes. 

The marketing of products, particularly supplements, foods and online programs, was pervasive throughout Instagram posts. This finding is not surprising, given that the estimated worth of the social media advertising industry is USD 269 billion and advertising on social media is now ubiquitous [41]. Furthermore, marketing executives in the wellness industry now use social media as one of their main forms of advertising [42]. Posts marketing supplements or online subscription services frequently made claims about health benefits and individuals viewing this content may be at risk of being misinformed. Health misinformation on social media is widespread, particularly in relation to nutrition [43], and marketing claims may be inaccurate or misleading. For example, posts outlined the benefits of vitamin C and protein supplementation; however, in Australia, these nutrients are generally consumed in sufficient quantities from foods in the diet and supplementation is unlikely to have additional benefits [44]. Moreover, a small study found that Instagram posts advertising supplements made misleading claims about “immune boosting” benefits [45]. Healthy food, such as fruit and vegetables, were also frequently advertised, which is a promising finding, given that the marketing of discretionary foods is also common on social media [46]. However, it is unclear from the data collected who is exposed to such marketing, and it is likely that individuals already interested in healthy eating are following or exposed to the marketing of healthy food due to the influence of the algorithm. 

The discussion of dieting to achieve weight loss and physique-related goals was prevalent throughout nutrition-related Instagram posts and tended to have very high engagement. This finding is supported by the large body of literature about the pervasiveness of weight loss content, dieting culture and the promotion of thin ideals on social media [47,48,49], and is evidence that Instagram content about weight loss receives high levels of engagement [15]. Dieting is a risk factor for the development of an eating disorder [50,51,52]. Furthermore, social media use and engagement with content that promotes weight loss and thinness is associated with body dissatisfaction and disordered eating [53,54]. The culture of dieting and thinness has received just scrutiny and there has been a recent transition on social media from the thin ideal toward a more athletic ideal as a healthier substitute, as seen in the rise of “fitspiration” content [47]. Fitspiration purports to educate and promote healthy behaviours, primarily related to exercise and diet [55]. Viewing fitspiration can contribute to poor body image and restrained eating [11,12,14,55]. In the present study, messages about food and nutrition were interwoven with messages about image. The frequent use of the term “body transformation” suggests that posts were focused on image as the primary motivator for dietary change, rather than health. There was often acknowledgement of the relationship between dieting and disordered eating in posts that promoted online weight loss services, which was contradictory of the posts’ underlying messages about dieting to achieve weight-related goals. Nutrition-related Instagram content may be contributing to the culture of unattainable beauty standards and individuals who engage with nutrition-related content may be at risk of body dissatisfaction and disordered eating; however, more research is required to confirm this. 

It is also important to note the impact of the COVID-19 pandemic on this study’s findings. The least dominant theme, “home”, captured posts that discussed carrying out tasks such as cooking and exercising at home and some posts discussed COVID-19 and the impact of lockdowns. Data for this study were collected across 2020 and 2021. During this period, different geographical locations in Australia were subject to strict lockdowns and work from home orders, and it is evident that this had an impact on the data collected. The pandemic saw a rise in home cooking due to changes in food supply and availability and a focus on home-based activities and pastimes [56,57], and people took to social media to share recipes and advice for baking “lockdown sourdough” [58]. Recipes and home-cooking content featured heavily throughout the sample and was captured under all themes. It is unclear if the prominence of home-cooking content would have been the same if the data were collected at a different point in time. Future studies on nutrition-related Instagram content that analyses posts from before, during and after the pandemic are required to confirm the impact of lockdowns on nutrition-related Instagram content. 

This is the first content analysis of nutrition-related information published by a large sample of influential Instagram accounts, and this study has several key strengths. Firstly, the large sample size of Instagram posts improves the generalisability of the data. The 12-month period of observation is another strength as seasonal differences in the Instagram content are captured, for example, the discussion of food indulgence during festive periods. Finally, the use of Leximancer to aid in the development of concepts and themes may have minimised the impact of researcher bias and provides an innovative approach to content analysis. This study also has a number of limitations that should be considered. Firstly, video content has expanded in popularity on social media; however, posts containing videos were excluded from the sample because it was beyond the scope of this study to transcribe them for analysis in Leximancer. Secondly, the content of images was not analysed; therefore, there may be some contextual information that was not captured. Thirdly, all content was authored by Australian brands or influencers and may not be generalisable in a global context. Fourthly, the impact of bots on follower counts and engagement data cannot be accounted for due to the sophistication of bots and limited data that Instagram allows access to. As such, it is possible that follower counts and engagement data were overestimated, which is a common limitation in social media research. Finally, the list of Instagram accounts that was screened were the top accounts listed under the “health” category on StarNgage’s website. Therefore, there may be accounts that post recipes but are not focused on health that were not captured in the sample. However, the impact of this may have been minimised through the snowballing technique that was used, and some accounts that posted recipes but were not health-focused were included. Due to the restriction of access to Instagram’s API, it is not possible to determine if the sample of accounts is comprehensive. More broadly, restricted access to social media platforms’ APIs is a significant challenge for researchers collecting social media data.

Findings from this study have implications for policy, practice, and future research. The popularity of healthy recipes and nutrition information observed in this study indicates that social media may be a useful setting for wide-reaching nutrition promotion campaigns. There is also scope for social media platforms to aid in health-promotion efforts by regulating content. Instagram and Facebook have banned the term “thinspiration” and the promotion of “skinny teas” and appetite-suppressing lollypops [59,60]. Similar regulation may also be useful for nutrition-related marketing content, for example, banning the promotion of junk food, regulating the claims that can be made about supplements and weight loss programs or mandating the inclusion of disclosure statements, to reduce the risk of misleading consumers. External regulatory bodies, such as Australia’s Therapeutic Goods Administration (TGA), are unable to scrutinise the volume of posts about products, and social media platforms should take greater responsibility for moderating the content published on their platforms. From July 2022, after the data collection period of this study, the TGA legislated against the influencer promotion of health products including supplements, and future research should investigate the impact of this legislation on social media content. Future research should investigate the quality and accuracy of nutrition-related Instagram content, and the impact that confusing or inaccurate information can have on nutrition knowledge and beliefs. Currently, little is known about who engages with nutrition-related social media content or how it informs food choice. A better understanding of social media’s influence on food choice may help inform successful social-media-based health-promotion campaigns, and therefore, warrants further investigation. 

## 5. Conclusions

This novel content analysis of 10,964 Instagram posts about nutrition found that Instagram content containing healthy recipes, nutrition information and advice about preparing healthy food was prevalent. Moreover, marketing for products such as supplements, foods and online subscriptions to meal plans and coaching programs was ubiquitous. Finally, there was a large amount of nutrition-related Instagram content that focused on weight loss and achieving physique-related goals. The popularity of content about nutrition and healthy eating indicates that social media could potentially be a useful health-promotion tool, and future research should explore the effectiveness of social media to promote nutrition. Conversely, the pervasiveness of marketing weight- and physique-focused nutrition-related content suggests that consumers may be exposed to misleading or potentially harmful information, and future studies should investigate the accuracy and impact of nutrition-related Instagram content.

## Figures and Tables

**Figure 1 nutrients-15-02332-f001:**
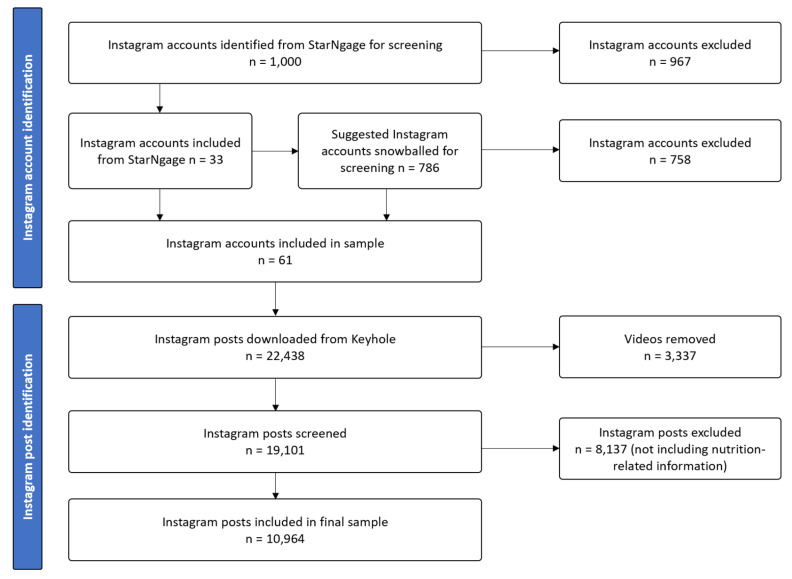
Flow chart of Instagram accounts and Instagram post sample selection.

**Figure 2 nutrients-15-02332-f002:**
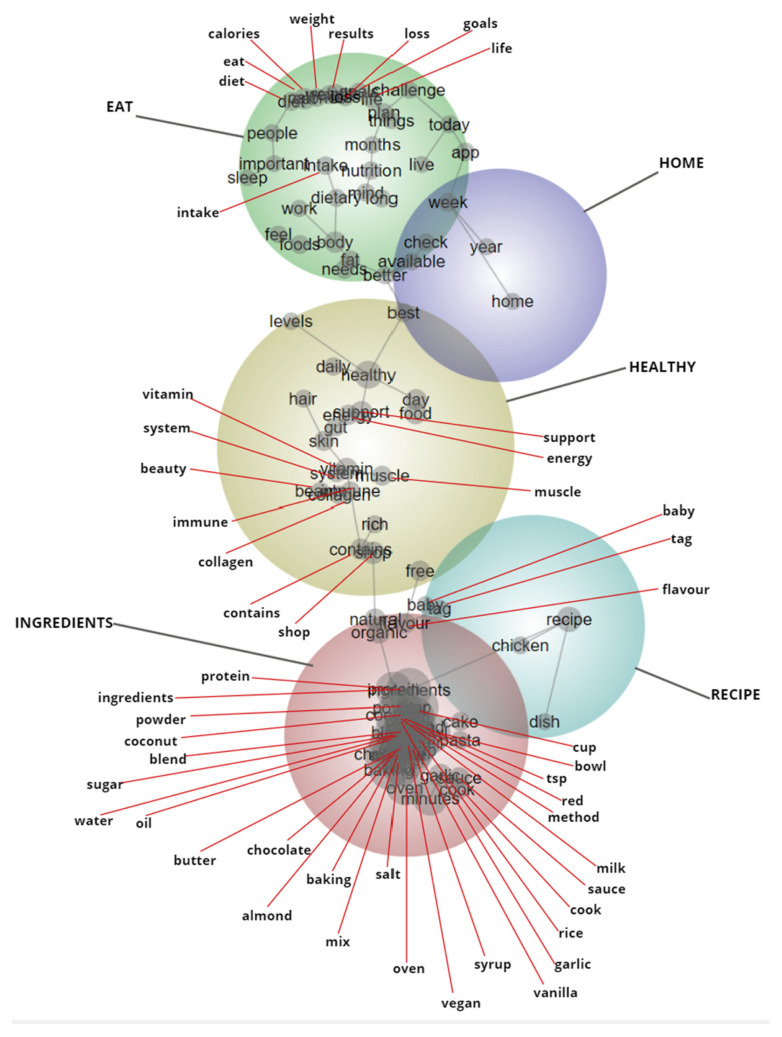
Concept map generated by Leximancer including all posts (*n* = 10,964). Themes are heat mapped where warmer colours indicate greater importance; theme names are indicated in capital letters and concepts in lower case.

**Table 1 nutrients-15-02332-t001:** Number of followers, engagement and posts for Instagram accounts included in the sample.

Instagram Account Category	Followers (Mean)	Followers (Standard Deviation)	Engagement * (Mean)	Engagement * (Standard Deviation)	Total Posts in Sample (*n*)
All (*n* = 61)	279,505	267,541	1854	3704	10,964
Brand (*n* = 23)	228,599	104,240	885	1778	5698
Media entity (*n* = 5)	293,338	63,287	1219	1037	1850
Recipes/meal ideas (*n* = 11)	256,212	106,207	2452	2512	1681
Fitness/coaching influencer (*n* = 8)	448,946	266,285	4493	6612	749
Nutritionist or dietitian influencer (*n* = 8)	252,589	215,935	3880	7054	634
Lifestyle influencer (*n* = 6)	275,773	161,766	2979	4387	352

* Engagement was calculated as the number of likes plus the number of comments per post for all posts across the 12-month period of observation.

**Table 2 nutrients-15-02332-t002:** Themes and concepts from all Instagram posts (*n* = 10,964) identified by Leximancer and engagement data for posts tagged with each concept.

Theme	Hits	Engagement (Mean ± Standard Deviation)	Concept	Count	Relevance
Ingredients * Recipes **	14,289	1382 ± 2061	ingredients	7265	100%
			cup	6140	85%
			bowl	5783	82%
			coconut	5690	78%
			oil	5324	73%
			mix	5298	73%
			powder	5263	72%
			method	5180	71%
			tsp	5150	71%
			baking	4991	69%
			milk	4904	67%
			minutes	4875	67%
			vanilla	4871	67%
			butter	4797	66%
			protein	4794	66%
			chocolate	4712	65%
			syrup	4540	62%
			almond	4338	60%
			salt	4322	59%
			oven	4140	57%
			blend	3767	52%
			water	3725	51%
			vegan	2905	40%
			sugar	2858	39%
			garlic	1769	24%
			cook	1741	24%
			flavour	1718	24%
			natural	1503	21%
			organic	1398	19%
			rice	1279	18%
			cake	1036	14%
			pasta	781	11%
			red	755	10%
Healthy * Food and nutrition practices **	12,051	1053 ± 2102	healthy	5182	71%
			food	2736	38%
			day	2533	35%
			shop	2449	34%
			support	2123	29%
			vitamin	1963	27%
			free	1769	24%
			energy	1612	22%
			skin	1519	21%
			contains	1451	20%
			muscle	1353	19%
			best	1257	17%
			daily	1015	14%
			beauty	953	13%
			gut	937	13%
			collagen	890	12%
			immune	870	12%
			hair	818	11%
			system	711	10%
			rich	673	9%
			levels	575	8%
Eat * Body goals **	11,401	2276 ± 3811	eat	3061	42%
			body	2219	31%
			week	1652	23%
			app	1508	21%
			results	1434	20%
			foods	1391	19%
			diet	1378	19%
			weight	1291	18%
			fat	1288	18%
			life	1248	17%
			today	1217	17%
			calories	1209	17%
			needs	1076	15%
			intake	976	13%
			better	950	13%
			nutrition	937	13%
			feel	919	13%
			check	906	12%
			plan	901	12%
			loss	824	11%
			goals	751	10%
			long	748	10%
			dietary	745	10%
			challenge	721	10%
			people	719	10%
			mind	705	10%
			work	689	9%
			months	640	9%
			available	613	8%
			things	588	8%
			important	549	8%
			live	425	6%
			sleep	371	5%
Recipe * Food literacy **	7859	1542 ± 1863	recipe	6287	87%
			tag	953	13%
			chicken	889	12%
			dish	535	7%
			baby	470	6%
Home *	1047	1969 ± 3276	home	563	8%
			year	518	7%

* Identified by Leximancer, ** name assigned by authors.

## Data Availability

The data presented in this study are available on request from the corresponding author. The data are not publicly available to protect the privacy of the Instagram accounts included in this research.

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
