# Peer review of "Nutrition-Related Information on Instagram: A Content Analysis of Posts by Popular Australian Accounts"

_nutrients, 2023, doi:10.3390/nu15102332_

Round 1

Reviewer 1 Report

Nowadays, almost everyone has an account on social media, including Instagram. On the one hand, it is a great tool for gaining knowledge about current events, peeking into your own authorities and developing passion. Social media is also, unfortunately, the cradle of self-proclaimed authorities in various fields, often not supported by Evidence Base Nutrition. Unfortunately, to be treated as an authority, it is enough to have a few thousand followers on Social Media. This can be seen, for example, in the huge increase in popularity of various diets, when the celebrity mentions their advantages. There are probably hundreds of thousands of accounts on Instagram related to diet and healthy eating, but most of them are not worth following. Many of them replicate dietary myths and encourage practices that have long been refuted by science. Research on content selection therefore seems particularly important. I read the article with great interest and I see that it was very well thought out and written, I have some doubts, in fact, which are more to satisfy my curiosity than affect the structure of the work:

#1 The choice of presented methods may involve some bias, which should be demonstrated in the limitations of the study

#2 Why only accounts with posts were selected, while rollers now dominate social media and allow you to get more followers

#3 What were the reasons for the most common misunderstandings at the account selection stage

#4 Customize the tables according to the editorial requirements of the magazine

#5 I also lack a clear emphasis that Instagram can promote content on eating disorders and affect body image and body image perception.

Congratulations on a very good article. Your research is well organized, insightful and makes a valuable contribution to this field. I see you put a lot of effort into your work, as you can see in your high-quality writing and analysis. Overall, I believe your article has the potential to make a significant impact on your field and I am confident it will be well received by the scientific community. Most importantly, it has practical implications that I hope will translate into national health policy.

Greetings!

Reviewer 2 Report

Thank you very much for allowing me to review the work entitled "Nutrition-related information on Instagram: A content analysis of posts by popular Australian accounts" (nutrients-2368668), which is presented for the "Nutrition and Public Health" section in the Special Issue "Digital Food Literacy, Tailored Nutrition, and Food Environment".

The qualitative study, which followed the guidelines set out by Elo et al., focuses on the new way of life in which social media has become very prominent, particularly on Instagram. The aim of this study was to examine the content of nutrition-related posts from popular Australian Instagram accounts in Australia between September 2020 to 2021.

I believe that this is a new approach to a growing topic, which is the power of media on behaviour, specifically in nutrition. Therefore, it is important for the methodology to be clearly explained. I suggest including the reasons that led to the extraction of the concepts described in the tables and a description of how the graphs were created in the methodology.

In the discussion, the strengths of the study should be addressed, such as its innovation, but also its limitations, such as it being conducted in Australia, which implies a certain economic, social, and cultural level, making the results not always applicable globally, especially since access to social media is not always possible in other countries.

The conclusions should be based on the results of the study.
